# Ion selectivity and rotor coupling of the *Vibrio* flagellar sodium-driven stator unit

Haidai Hu[1], Philipp F. Popp[2], Mònica Santiveri [1], Aritz Roa-Eguiara [1], Yumeng Yan[1], Freddie J. O. Martin [1], Zheyi Liu[3,4], Navish Wadhwa [5,6], Yong Wang [3,4], Marc Erhardt [2,7] & Nicholas M. I. Taylor [1] ✉

Bacteria swim using a flagellar motor that is powered by stator units. *Vibrio* spp. are highly motile bacteria responsible for various human diseases, the polar flagella of which are exclusively driven by sodium-dependent stator units (PomAB). However, how ion selectivity is attained, how ion transport triggers the directional rotation of the stator unit, and how the stator unit is incorporated into the flagellar rotor remained largely unclear. Here, we have determined by cryo-electron microscopy the structure of *Vibrio* PomAB. The electrostatic potential map uncovers sodium binding sites, which together with functional experiments and molecular dynamics simulations, reveal a mechanism for ion translocation and selectivity. Bulky hydrophobic residues from PomA prime PomA for clockwise rotation. We propose that a dynamic helical motif in PomA regulates the distance between PomA subunit cytoplasmic domains, stator unit activation, and torque transmission. Together, our study provides mechanistic insights for understanding ion selectivity and rotor incorporation of the stator unit of the bacterial flagellum.

Many bacteria rotate flagella to power their movement. The flagellum is characterized by a long filament, connected through a flexible hook to the cell envelope embedded rotary motor (or basal body), which comprises a rotor and multiple stator units[1–4]. The flagellar stator unit uses the transmembrane ion motive force (IMF) to generate mechanical torque to rotate the flagellum, which is employed by many bacteria to direct their locomotion in liquid environment or on viscous surfaces to a favorable niche[4–6]. Driven by the stator unit, the bacterial flagellar motor can rotate in both clockwise (CW) and counterclockwise (CCW) directions, with the switch between the two directions controlled by intracellular chemotaxis signaling[7,8]. The stator units are strictly required for rotation of the flagellum and thus motility of the bacteria, but not for flagellar assembly[9,10]. In addition, the stator units dynamically associate with and dissociate from the rotor[11–13]. Changing the number of engaged stator units allows tuning the required torque in relation to the mechanical load[14–18].

Each stator unit is composed of two membrane proteins assembled as a complex buried inside the cytoplasmic membrane, in which their transmembrane domains organize as an ion channel[19,20]. Incorporation of the stator unit requires its cytoplasmic domain to interact with the rotor and its periplasmic domain to attach to the bacterial cell wall[21]. Torque generated by ion translocation is transmitted to the rotor via electrostatic interactions at the stator-rotor interface[22–25]. Depending on the conducting ions, stator units can be mainly grouped into two subfamilies: the H⁺-driven stator unit (e.g., MotAB) and the Na⁺-driven stator unit (e.g., PomAB)[26,27]. In addition, stator units using potassium and divalent ions such as calcium or magnesium as coupling ions have also been reported[28–31]. Recently, single particle cryo-

[1]Structural Biology of Molecular Machines Group, Protein Structure & Function Program, Novo Nordisk Foundation Center for Protein Research, Faculty of Health and Medical Sciences, University of Copenhagen, Blegdamsvej 3B, 2200 Copenhagen, Denmark. [2]Institute for Biology/Molecular Microbiology, Humboldt-Universität zu Berlin, Philippstr. 13, 10115 Berlin, Germany. [3]College of Life Sciences, Zhejiang University, Hangzhou 310027, China. [4]The Provincial International Science and Technology Cooperation Base on Engineering Biology, International Campus of Zhejiang University, Haining 314400, China. [5]Department of Physics, Arizona State University, Tempe, AZ 85287, USA. [6]Biodesign Center for Mechanisms of Evolution, Arizona State University, Tempe, AZ 85287, USA. [7]Max Planck Unit for the Science of Pathogens, Berlin, Germany. ✉e-mail: nicholas.taylor@cpr.ku.dk

electron microscopic (cryo-EM) structures of H⁺-driven MotAB stator units[32,33], cryo-EM structures of intact flagellar motor complexes[34–36], as well as in situ cryo-electron tomographic (cryo-ET) studies of the flagellar motor[21,37–40], provided detailed structural and functional views of stator unit assembly, torque generation and motor function[1]. The data strongly suggest a rotational model for the mechanism of action of the stator units. Upon dispersion of the IMF, MotA is proposed to rotate around MotB, which is anchored to the peptidoglycan layer. By engaging with the rotor, MotA rotation powers the rotation of the large rotor. A differential engagement of MotA with the rotor between the CW and CCW states of the rotor is proposed to form the mechanistic basis of switching.

The Na⁺-driven stator unit is particularly important for *Vibrio* species, including pathogenetic ones (e.g., *V. cholerae*, *V. alginolyticus*), as their polar flagella can only be powered by the transmembrane Na⁺ gradient, and the motility of many *Vibrios* has been linked to their virulence, biofilm formation and dispersion[41–43]. However, at the molecular level, how stator units discriminate among different types of ions and power rotation of the flagellar motor have remained unclear. Furthermore, the Na⁺-driven stator unit is an ideal subject for investigating stator unit ion selectivity and translocation mechanisms. As a Na⁺ ion interacts more with electrons than a proton in the cryo-electron microscope, it could potentially be visualized more readily in a high-resolution cryo-EM map. Finally, sodium ions are easier to be detected and manipulated than protons[44].

An atomic structure of the Na⁺-driven stator unit is thus crucial for elucidating the mechanism of how the stator unit distinguishes ions and couples ion transportation into its rotation. To this end, we determined cryo-EM structures of *Vibrio* PomAB (from *V. alginolyticus*; termed as *Va*PomAB) in both detergent and lipidic environments, with the local map resolution reaching up to ~2 Å. The high-resolution structure enabled us to locate Na⁺ ion binding sites and revealed the structural and mechanistic basis of ion selectivity. We show at the molecular level how the stator unit achieves its monodirectional rotation upon ion transport. Furthermore, we identified a helical motif in the C-terminal region of PomA (CH) that is essential for stator unit function. We validated our structural results through extensive mutagenetic analysis and molecular dynamics (MD) simulations. Finally, we propose a role for the asymmetric cytoplasmic domain arrangement of the stator unit in torque generation and the assembly/disassembly mechanism of the stator unit into the rotor.

## Results

### Structure determination and *Va*PomAB architecture

Intact *Va*PomAB is an anisotropically shaped complex and shows preferential orientation of particles in vitreous ice (Fig. 1, Supplementary Fig. 2). To improve sample homogeneity, we modified the protein purification protocol and encoded a protease site in the PomB gene after the plug region, which allowed for the removal of the PomB peptidoglycan binding domain (PGB) during protein purification

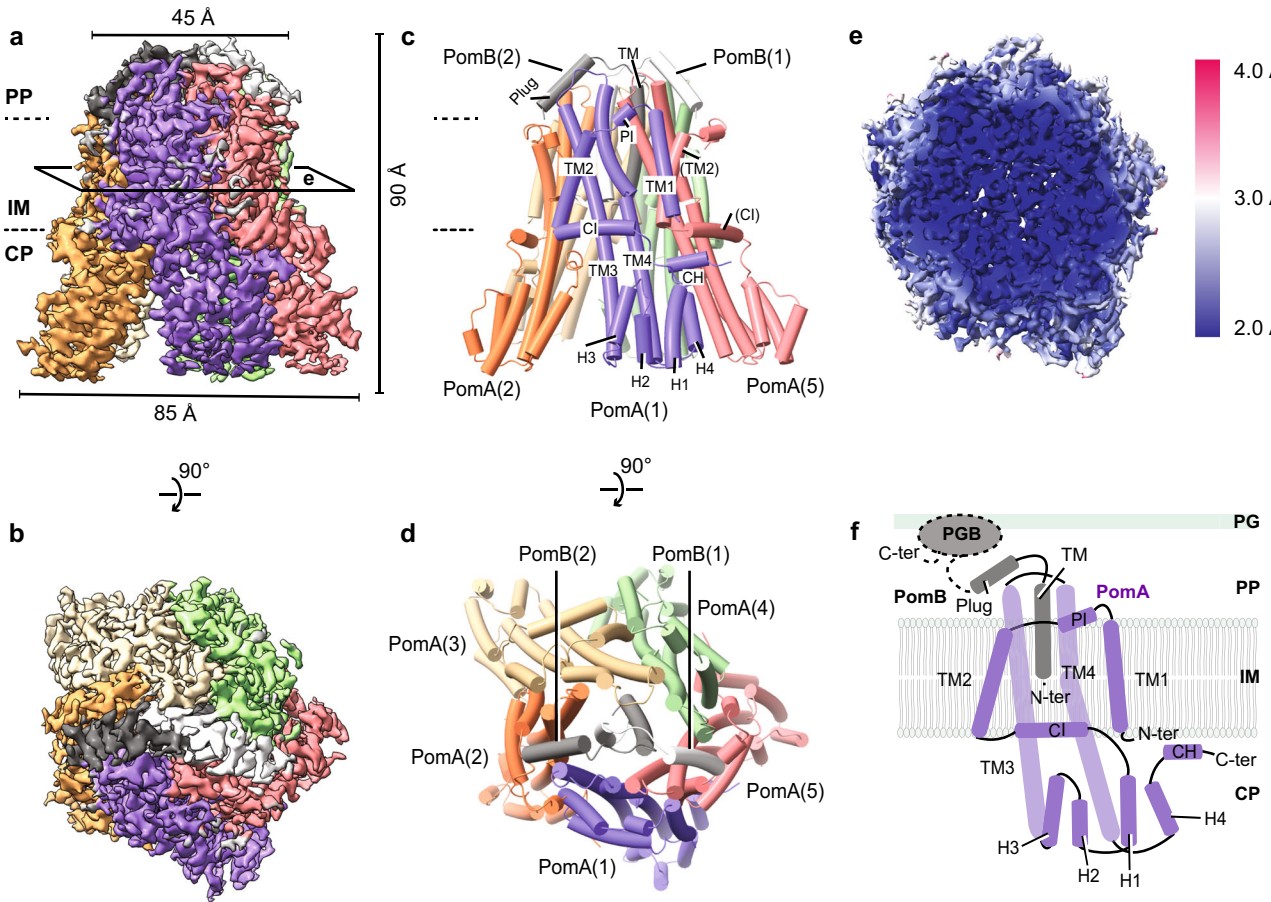

**Fig. 1 | Cryo-EM map and overall architecture of the Na⁺-driven stator unit *Va*PomAB. a** Cryo-EM map of *Va*PomAB. PomA subunits (purple, orange, yellow, green and red) surround PomB subunits (black and white) viewed from the plane of the membrane. Dashed lines represent approximate inner membrane boundaries. **b** Cryo-EM map of *Va*PomAB viewed from the periplasmic side. **c** Ribbon model representation of *Va*PomAB. Subunits are colored as in (**a**). **d** *Va*PomAB model viewed from the periplasmic side. **e** Local resolution map of *Va*PomAB viewed from a cross section as indicated in (**a**). **f** Topology diagram and secondary structural elements of *Va*PomA (purple) and *Va*PomB (black) subunits. The gray ellipse indicates the PomB peptidoglycan-binding domain (PGB). PP periplasm, IM inner membrane, CP cytoplasm, PG peptidoglycan, TM transmembrane, H helix.

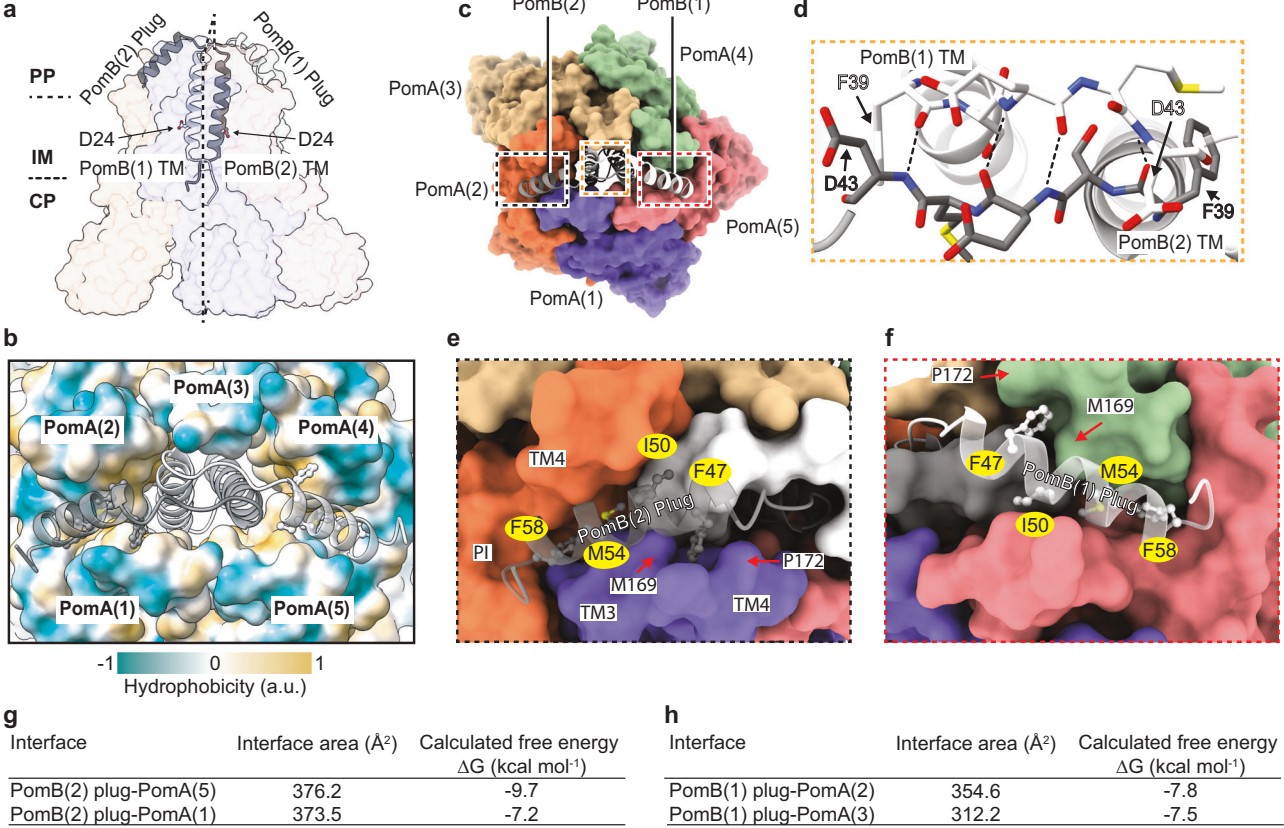

**Fig. 2 | PomB plug motif and auto-inhibition mechanism. a** *Va*PomAB in its auto-inhibited state, viewed from the plane of the membrane, with PomB shown as ribbons (black and white) and PomA shown as a semitransparent surface representation. The aspartate residues D24 from both PomB TM are indicated and shown as sticks. **b** Top view of *Va*PomAB with PomB shown as ribbons and PomA shown as a surface representation colored according to its hydrophobicity. **c** Top view of *Va*PomAB. PomA subunits are shown as a surface representation and PomB subunits are displayed as ribbons, colored as in Fig. 1a. **d** Close-up view from the periplasmic side of the interactions of the linkers (Phe39-Asp43) that connect PomB plug motifs and TMs (it corresponds to the yellow box in (**c**)). Hydrogen bonds are represented as dashed lines. **e** Plug motif from PomB(2) binding environment (black box in (**c**)). **f** Plug motif from PomB(1) binding environment (red box in (**c**)). **g, h** Calculated interface buried area and free energy of PomB plug motifs.

(Fig. 1, Supplementary Fig. 1b). To overcome the preferred orientation, we added the zwitterionic detergent CHAPSO to randomize particle orientation during EM grid preparation[45,46]. Single-particle analysis yielded an overall resolution of *Va*PomAB in Lauryl Maltose Neopentyl Glycol (LMNG) detergent of ~2.5 Å, with the cryo-EM map of sufficient quality to build an atomic model for most of the protein complex. The local resolution corresponding to the inner transmembrane domain approaches 2 Å, with clear density for non-protein molecules, allowing us to model water molecules and ions, as well as residue side chains isomers (Fig. 1, Supplementary Figs. 2–4 and Supplementary Table 1).

PomAB assembles the characteristic bell shape of the stator unit family, with conserved 5:2 subunit stoichiometry and overall architecture. Five PomA molecules arrange pseudo-symmetrically around two PomB, with each PomA subunit comprising four transmembrane helices (TM1-TM4) folded into two radial layers. The TM3s and TM4s form an inner layer lining the dimerized PomB TMs. The TM1s and TM2s surround peripherally, together with PomA periplasmic interface (PI) helices and cytoplasmic interface (CI) helices, establishing an outer layer, with one side packing against the inner layer and the other side hydrophobically interacting with the lipid bilayer. The resolved TM1 of one PomA makes prominent contact with the TM2 from the adjacent subunit. The cytoplasmic domain of PomA contains a compact helix bundle (H1-H4), a region where torque is generated through electrostatic matching with the rotor FliG torque helix[47]. The cryo-EM map of PomAB also reveals a short helix after PomA H4, which we designated as CH (C-terminal helix) motif, attaching to the CI helix of a

neighboring PomA subunit (Fig. 1f). The plugged motifs from two PomB chains are fully resolved in our PomAB structure, where they are positioned on the top of the periplasmic side of the stator unit, consistent with a plugged autoinhibited state. We also noticed that each end of the plug motif interacts with the PI helix of PomA. We propose this causes the N-terminal residues (residues 1–21) of two PomA subunits to be disordered, as these are not resolved in our cryo-EM map (Supplementary Fig. 5c).

**Plug motif and autoinhibition mechanism**

The PomB plug motif is a short amphipathic α-helix, following the TM helix. Deletion of the plug motif results in ion influx into the cell cytosol, and thus causes cell growth inhibition when overexpressed[48,49]. Earlier studies through mutagenesis and cross-linking experiments have mapped critical residues involved in interactions between the PomB plug motif and PomA[48,50]. In the PomAB structure, the TM of PomB is connected to the plug helix through a four-residue linker (Fig. 2c), which makes the plug helix turn ~145°, rendering its C-terminus to point towards the cytoplasmic membrane. The two short linkers establish interaction laterally by four backbone hydrogen bonds, organizing the plug motifs as a trans-mode configuration relative to PomB TM helices, with a pseudo-mirror symmetry perpendicular to the cell membrane (Fig. 2a, d). The plug motifs from sodium and proton-driven stator units share a similar amino acid pattern (Supplementary Fig. 1b) comprising a hydrophobic side that makes its main interaction with the stator unit itself and a hydrophilic

side that is exposed in most parts to the periplasmic space solvent (Fig. 2b). The contact environment of the PomB plug motif is mainly contributed by a cleft framed by the periplasmic side of the TM4, TM3 and the PI helix from one PomA subunit, and the periplasmic side of the TM3 and TM4 from the adjacent PomA subunit. Three residues from the plug motif (I50, M54 and F58) deeply insert into this cleft, establishing hydrophobic interactions (Fig. 2e, f). In addition, the PomB F47 aromatic ring is sandwiched between the pyrrolidine ring of P172 and the side chain of M169 from PomA, via CH-π interactions, further stabilizing the plug motif (Fig. 2f). The 5:2 stoichiometry of the stator unit creates inequivalent binding environments for the two plug motifs, as examined by calculating the surface buried area and free energies of residues forming the plug helix (residues 44–58, Fig. 2g, h). Therefore, we speculate that during stator unit activation, releasing the plug motif from the stator unit is not a symmetric process. Instead, one plug motif with relatively low binding energy likely detaches from its inhibitory site first, and the second plug motif will then be induced to be released. PomB G59 marks the end of the plug motif, and it directly exerts the effect on the conformation of PomA PI helix. We found that each of the PomB plug motifs induces two different conformations of the PomA PI helix that links PomA TM1 and TM2; one conformation is akin to those observed in the other three PomA subunits, and the other conformation extends TM2 one more helical turn involving residues from L26 to V32 (Supplementary Fig. 5c).

The dynamics of the PomA PI helices stemming from the PomB plug motif interaction presumably drives the flexibility of the corresponding TM1, as the latter could not be resolved in two of the PomA subunits. The high-resolution PomAB structure was determined in a detergent micelle environment, raising the possibility that detergent molecules could have an impact on the conformation of PomAB, particularly the membrane-facing helices, including the disordered TM1 from two PomA subunits. To clarify this and to better mimic the native environment of the stator unit, we reconstituted *Va*PomAB into membrane scaffold protein 1D1 (MSP1D1) nanodiscs, as well as full length, non-cleaved *Va*PomAB into saposin nanodiscs, with *E. coli* polar lipids, and determined the map resolution, at 3.9 Å and 6.3 Å, respectively (Supplementary Figs. 3 and 4). In both cases, we were able to trace all the secondary structure elements of the PomAB complex, except the two PomA TM1 helices (Supplementary Fig. 5f, 5i). Comparison of the PomAB LMNG structure to the MSP1D1 lipid-reconstituted structure did not reveal major conformational differences (root-mean-square deviation of 0.642 Å, 1,312 $C_\alpha$ atoms aligned) that could arise from detergent artifacts. This indicates that the flexibility of those two TM1 helices in the inactive stator unit is probably intrinsic, which might be functionally important during stator unit activation.

## Na⁺ ion binding sites and ion selectivity mechanism

Stator units use specific ions to power flagellar motor rotation. Each MotB/PomB TM contains an aspartate (D24 in PomB) that is responsible for the binding and translocation of incoming ions from the periplasmic space to the cytoplasmic side (Fig. 2a). However, this aspartate is universally conserved among stator unit families (Supplementary Fig. 1b), obscuring the structural and mechanistic basis of the ion selectivity. The PomAB structure shows that the D24 of two PomB chains sit in different environments; D24 of Pom B chain 1 interacts with PomA, which we refer to as an engaged state; while D24 of PomB chain 2 points towards the cytoplasmic domain and breaks the interaction with PomA, which we refer to as a disengaged state (Fig. 3a). Examination of the high-resolution density map in the vicinities of these two aspartates reveals nonresidue densities. In site 1, close to the engaged PomB D24 (PomB chain1), the extra density is coordinated by oxygens from side chain hydroxyl groups of PomB T21 and D24, and backbone carbonyl groups of adjacent PomA P151 and PomB G20. A fifth coordinating interaction is made by a hydrogen

bond from a water molecule near PomA A190, and the average distance between the center of the density and associated oxygens is 2.88 Å (Fig. 3b). In site 2, near the disengaged PomB D24, which is more flexible as indicated by the slightly blurred EM density of its acidic side chain, a globular density is well coordinated by oxygen atoms exclusively contributed by PomA TM3 and TM4: side chain hydroxyl groups of T158, T185 and T186, and exposed backbone carbonyl groups of G154 and A182, with an average distance between the density center and associated oxygen of 2.33 Å (Fig. 3c). Given the cation's favorable local chemical environments in these two sites, and especially the typical geometry of Na⁺ coordination[51] in site 2, we modeled these densities as Na⁺ ions, which were the most predominant cations in the protein purification buffer. To further validate the model, we performed two explicit solvent all-atom MD simulations (1 μs for each) and observed that the Na⁺ ion in site 1 was very stable, but the other Na⁺ ion in site 2 rapidly moved to an intermediate site formed by the side chain of D24, T158, and T186 and subsequentially to a location symmetric to site 1, and finally released to the cytoplasmic space (Supplementary Fig. 6c–d and Supplementary Movie 1). We also observed significant conformational dynamics of a few polar residues around site 2, especially T158, T186 in PomA chain 5 and D24 in PomB chain 2 (Supplementary Figs. 6a and 7). By contrast, T186 in PomA chain 2 and D24 in PomB chain 1 on the engaged site were however much more stable (Supplementary Figs. 6a–b and 7d).

The identification of the Na⁺ binding sites from EM density and the asymmetric conformational dynamics led us to speculate that at least part of the PomAB ion selectivity filter nests within the PomA subunit, and those three threonine residues (PomA T158, T185, and T186), which are conserved in all sodium-driven stator units (Supplementary Fig. 1a), account for the Na⁺ ion selectivity and transportation. Of note, the T158 from PomA chain 2, near the engaged PomB D24, does not directly contribute to the Na⁺ binding. Instead, it orients its side chain to establish a hydrogen bond with PomB D24 (Fig. 3b), indicating that a local conformational change occurs during Na⁺ ion transportation. Similarly, on the same site of the other three PomA subunits, we did not observe densities corresponding to a Na⁺ ion (Supplementary Fig. 8a–f), suggesting only one Na⁺ ion would be supplied during each stator unit rotational step.

To probe the functional role of key residues for ion selectivity of the Na⁺-driven stator unit, we first designed a chimeric PomAB (renamed as *Va*PotAB) by replacing PomB PGB with *S. enterica* MotB PGB, a strategy similar to that used in previous studies[52,53]. A plasmid encoding *Va*PotAB conferred a motile phenotype on soft agar plates when transformed into a mutant *Salmonella enterica* strain that lacks its wildtype MotAB (Fig. 3i). We then generated point mutations in *Va*PotAB to evaluate the significance of the three key threonines on flagellar motor rotation by examining the motility phenotype. We found that substituting any of these three threonines to alanines abolishes bacterial motility, confirming the importance of these residues for stator unit function (Fig. 3i). The Na⁺ ion binding cavity therefore seems a strict requirement for ion selectivity. We note that a K⁺ ion, which has a larger radius than Na⁺ ion (1.52 Å vs 1.16 Å) and has an average ligated bond distance of around 2.7–3.2 Å, cannot be accommodated in this cavity. To quantify the Na⁺ selectivity in this cavity, we performed free energy perturbation calculations and found that Na⁺ has a lower binding free energy than K⁺ by 4.37 ± 0.7 kJ/mol, indicating that this cavity prefers Na⁺ over K⁺. On the other hand, H⁺ is too small to fill this cavity, and it is energy unfavorable for a $H_3O^+$ to be liganded with a coordination number of five. Therefore, K⁺ and H⁺ (or $H_3O^+$) cannot be used by PomAB as coupling ions. Divalent ions, such as $Ca^{2+}$ and $Mg^{2+}$, which would need further negatively charged residues to be neutralized and coordinated, are likely not favored in this cavity either.

In addition, we compared the PomAB structure with the available H⁺-driven stator unit structures, *C. jejuni* MotAB and *B. subtilis* MotAB,

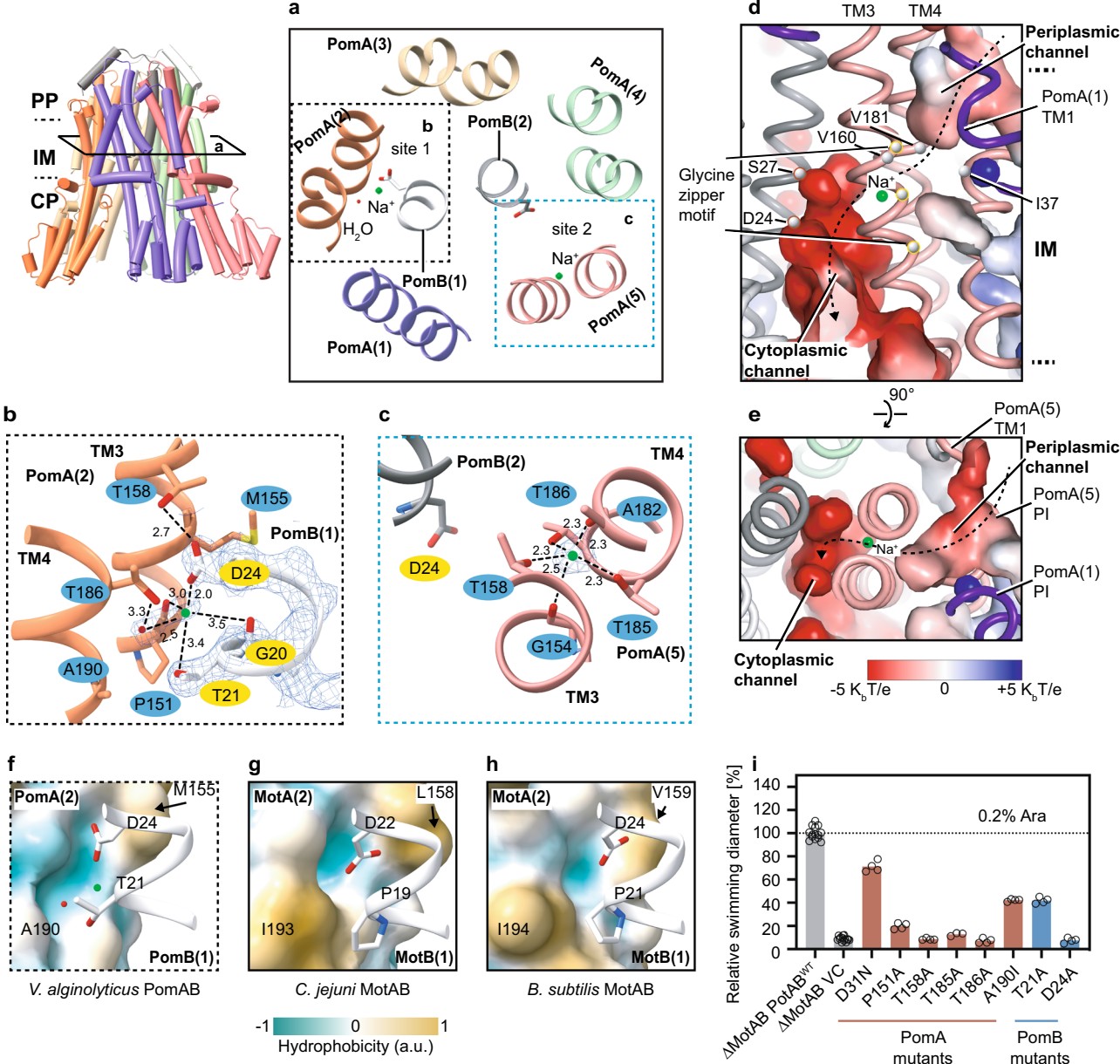

**Fig. 3 | Ion binding sites, selectivity, and translocation pathway. a** Cross section view (corresponding to the view in left panel and rotated 90°) of Na$^+$ ion binding sites (cyan spheres) in the vicinities of the two Asp24 from PomB. **b** Details of the Na$^+$ ion binding site near PomB(1) engaged Asp24. For clarity, corresponding EM densities are only overlapped in the region of PomB(1) Gly20-Asp24, Na$^+$ ion, and water molecule. Hydrogen bonds are indicated as dashed lines with distances in angstroms. **c** Details of the Na$^+$ ion binding site near disengaged PomB(2) Asp24. EM density is overlaid on the Na$^+$ ion. **d** Na$^+$ ion translocation pathway (dashed line with arrow). Periplasmic and cytoplasmic channels are indicated, with surface colored by electrostatic potential (positively charged, blue; negatively charged, red). Cα atoms of the residues forming the putative hydrophobic gate, of the glycines forming the glycine zipper motif, and of the PomB (2) S27 and D24 Cα are indicated and shown as spheres **e**, Top view of the Na$^+$ ion translocation pathway. **f** *Va*PomAB sodium ion binding environment near the engaged site. The surface of PomA is colored by hydrophobicity. **g** Similar view as in (**f**), but in the proton-driven stator unit *Cj*MotAB. **h** Similar view as in (**f**), but in the proton-driven stator unit *Bs*MotAB. **i** Comparison of motility ability of the *Va*PotAB constructs and point mutants of the residues near the Na$^+$ ion binding site or residues along Na$^+$ translocation pathway. "Δ*MotAB VC*" stands for an empty vector transformed into a mutant *Salmonella enterica* strain that lacks its wildtype MotAB. Source data are provided as a Source Data file.

to explore the reason why H$^+$-driven stator units cannot use sodium or other alkaline metals as coupling ions. In the part of the structure of the H$^+$-driven stator unit that is equivalent to the corresponding Na$^+$ binding site 2 in PomAB, two threonines (T158 and T185) are replaced by alanine, lacking oxygen in this cavity, likely precluding alkaline metal ion binding (Supplementary Figs. 1a and 8g–i). In the equivalent position of the PomB engaged D24, near the water molecule that coordinates the Na$^+$ binding site 1, the H$^+$-driven stator unit contains an isoleucine residue instead of an alanine or a polar residue, which makes

this region hydrophobic and does not favor an alkaline metal ion coordination (Fig. 3f–h). Thus, both sites in the H$^+$-driven stator unit lack the contact environment for alkaline metal ions, supporting the idea that the residue variability in PomA/MotA has a large influence on ion selectivity.

Analysis of the structure assembly interface between PomA and PomB subunits at the periplasmic level reveals that this inner contact interface is mainly lined by hydrophobic residues (Fig. 4a), with the thickness spanning around four helical turns (from PomB S27 to S38).

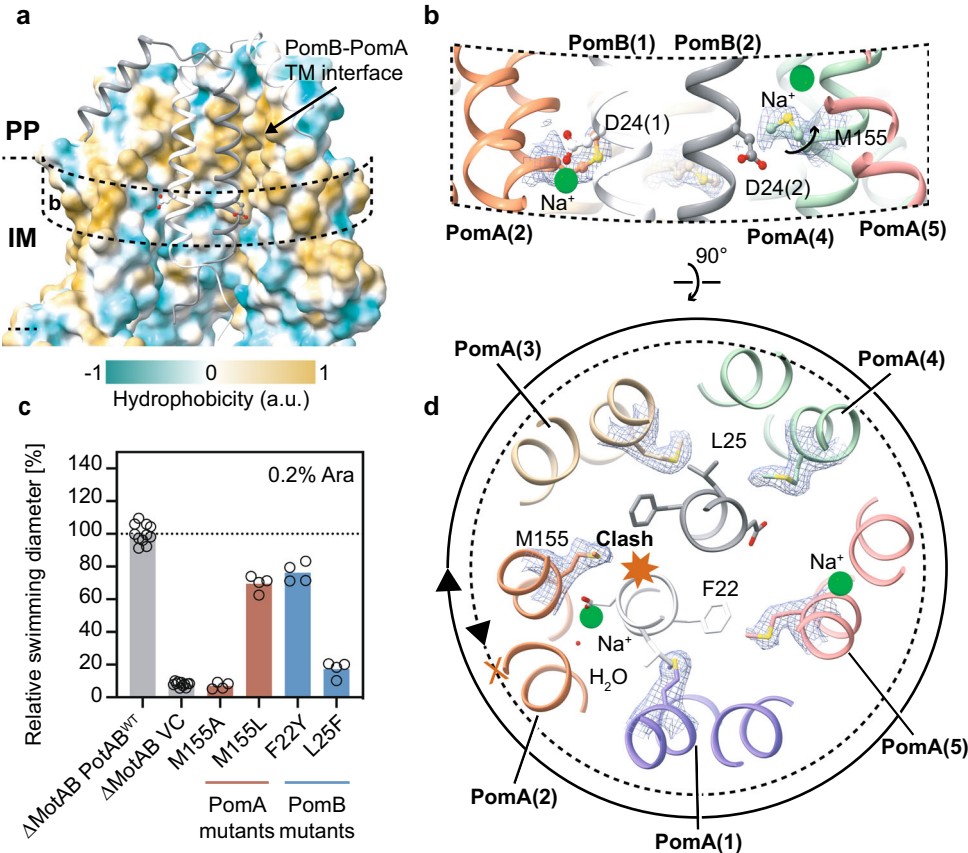

**Fig. 4 | VaPomAB assembly interface and its directional rotation. a** VaPomAB assembly interface at the periplasmic space and transmembrane domain levels, with surface colored according to hydrophobicity. For clarity, the front two chains are deleted and PomB chains are shown as ribbon. **b** Conformational isomers of M155 near PomB engaged D24 and disengaged D24. EM densities are overlaid on the side chain of M155. **c** Comparison of motility ability of the VaPotAB constructs and point mutants of the residue M155, and residues from PomB near M155. Source data are provided as a Source Data file. **d** Conformational isomers of M155 viewed from the top of the membrane. The solid circle indicates the rotational direction of PomA around PomB. A potential clash that would occur if PomA rotated CCW around PomB is indicated with a red heptagon.

It is therefore unlikely that an aqueous channel that mediates the $Na^+$ ion flow through PomAB is formed in this region. Rather, a potential $Na^+$ translocation pathway could be delineated based on the PomAB structure and our functional motility assay. It extends from the $Na^+$ binding site 2 to the periplasmic space, delineated on one side by the PI helix and the beginning of TM2 helix from the same PomA subunit and on the other side by the end of TM1 from the adjacent PomA subunit (Fig. 3d, e). The ion translocation pathway in this part contains a hydrophobic gate (Fig. 3d), likely removing the hydration shell of the incoming $Na^+$; and towards the periplasmic space, the translocation pathway is lined by several polar residues, such as D31, T33 and S34, and many of them are conserved (Supplementary Fig. 9b–c). The $Na^+$ translocation pathway reaches to the PomB D24 and to PomA cytoplasmic domain inner lumen, where the surface electrostatic potential is very negative (Fig. 4d), and, together with the N-terminus of PomB that harbors several negatively charged residues (Supplementary Fig. 1b), might attract the incoming $Na^+$. We also found that PomA TM3 contains a strictly conserved GXXGXXXG (residues G154-G161) motif, a typical 'glycine zipper' structure contributing to channel formation in many membrane proteins[54]. Glycines from the 'glycine zipper' motif face TM3 and TM4 assemble interface, holding the $Na^+$ selectivity filter in a middle position, and together with the conserved P151, contributing to the main chain conformational elasticity of this region when a $Na^+$ ion passes through TM3 and TM4 cleft. Substitution of the glycine residue to leucine in the 'glycine zipper' abolishes bacterial motility (Fig. 3d, Supplementary Figs. 1a and 14a).

From our explicit solvent MD simulations, we also observed that in the periplasmic side, the side chain of T33 was conformationally dynamic and surrounded by water molecules, which could occasionally diffuse to the space next to the side chain of T185 (Supplementary Fig. 7c and Supplementary Movie 2), therefore we propose that the hydration pocket form by T33 and a few other polar residues is the entry site of the proposed $Na^+$ translocation pathway. Note that we did not observe a continuous hydration or $Na^+$ translocation pathway to connect the periplasmic side and the $Na^+$ site 2, probably because this structure was in the self-inhibited plugged state and the simulation time (1 μs) was also much shorter than the timescale of channel opening.

### The stator unit is primed for directional rotation

Having analyzed the ion selectivity mechanism of the stator unit family in the context of the high-resolution map of PomAB, we next sought to understand the structural basis of the rotational direction of the stator unit. Cryo-ET studies in *V. alginolyticus* and *Borrelia burgdorferi* basal bodies reveal that when the rotor rotates in the CCW direction, its C-ring component FliG interacts with the stator unit cytoplasmic domain proximal side (the side facing the motor axis); while, when the rotor is locked and rotates in the CW direction, FliG interacts with stator unit cytoplasmic domain distal side. The rotor directional switching from CCW to CW rotation requires remodeling and expansion of the C-ring by changing its conformation upon receiving an intracellular chemotaxis signal[38,39]. Thus, the stator unit can drive both

CW and CCW rotation of the flagellar motor with a relatively fixed position by anchoring itself to the peptidoglycan layer through the PGB motif.

Viewed from the plane of the inner membrane, we observe that the bulky hydrophobic side chain of M155 from PomA chain 2 is oriented horizontally to the engaged PomB D24 (Fig. 4b), revealing that M155 will sterically hinder PomA to CCW rotation around PomB at the engaged D24 site (Fig. 4d). Meanwhile, M155 from PomA chain 4 elevates its side chain to stride over the disengaged D24, for which the interaction with PomA is nearly absent, providing the required space for D24 to gather the Na$^+$ ion from the selectivity cavity (Fig. 4b, d). We hypothesized that the bulky side chain of PomA position 155 is the stator unit directional rotation 'reinforcement' point. To test this hypothesis and verify the importance of the bulky side chain at this position, we first substituted this methionine residue with alanine. The M155A mutation abolished bacterial motility. In contrast, the replacement of methionine with leucine, a residue in the equivalent position often seen in H$^+$-driven stator units, retained 80% motility. Increasing the size of the residues near PomA M155 from PomB (PomB F22Y and L25F) impaired motility (Fig. 4c). Therefore, our structural analysis and functional data confirm that a residue with a bulky side chain near the ion coupling site (D24 in PomB) is required to permit the correct rotational direction of the stator unit. Its conformational isomer (Supplementary Fig. 10), likely induced by the local structural rearrangement during the stator unit activation, is necessary to achieve flexibility in this region for ion transportation. This bulky hydrophobic residue is conserved not only in flagellar stator units, but also in other 5:2 rotary motors[55], suggesting a similar directional rotation 'reinforcement' mechanism (Supplementary Fig. 11). The stator unit is thus a preset CW rotary motor, which is tightly blocked by the trans mode conformation of the PomB plug motif at the periplasmic side before it incorporates onto the rotor. The geometry of the stator unit will not favor a model where PomA rotates CCW around PomB, when the ion motive force is reversed, due to the structural clashes (Fig. 4d) and negative electrostatic potential of PomA cytoplasmic inner lumen. This is consistent with early experiments showing that the stator unit is inactivated when the IMF is dissipated or reversed[56], and that an increased sodium concentration in the cytoplasm inhibits the rotation of PomAB[49].

### PomA cytoplasmic domain and C-terminal helical motif

The stator unit cytoplasmic domain plays a crucial role during rotor incorporation, torque generation, and disassembly from the rotor[1,44]. The cytoplasmic domain of each PomA subunit contains four short helices that are almost vertical to the inner membrane. They peripherally surround the intracellular part of the PomA TM3 and TM4 helices and together form a compact helical bundle that protrudes ~35 Å into the cytosol (Fig. 5a–c). The cytoplasmic domains from five PomA subunits diverge towards their intracellular end, with the local resolution of this region decreasing considerably compared to the TMD. This is in line with the model B-factor distribution, where the PomAB cytoplasmic domain has a higher B-factor value (Supplementary Fig. 12), reflecting the flexibility of this region. The rotary stator unit generates torque by matching the complementary charged residues with the rotor FliG torque helix. This torque-generating mode is predicted to be conserved across bacterial species[22,24,57]. We define the FliG torque helix-binding interface from the stator unit as follows: positively charged residues from one PomA subunit contribute to the principal face or (+) face, and negatively charged residues from the neighboring PomA subunit mainly contribute to the complementary face or (−) face (Fig. 5b, c). The PomAB structure allows us to map the locations of those key residues involved in stator-rotor interaction. We found three positively charged residues from H1 and H4 at the (+) face, R88, K89, and R232, and two negatively charged resides from H2 and

H3 at the (−) face, D114 and E96, that when the charge is suppressed or reversed, greatly impair motility (Fig. 5h). Importantly, the charge of R88 at the (+) face and D114 and E96 at the (−) face, whose side chains project toward the PomA intersubunit junction, are indispensable for motility, confirming that both (+) and (−) sides of PomA are necessary and directly involved in the interactions with FliG torque helix. Besides, R232 establishes an interdomain salt bridge with residue D85, which is probably not only stabilizing helix bundle organization, but also contributing to the FliG torque helix binding. The R232A mutation impairs motility, and the D85A mutation results in a gain-of-function phenotype (Fig. 5b).

Unexpectedly, we found a helical (CH) motif right after the H4 helix in the PomA C-terminal part. The CH motif runs parallel to the membrane plane and attaches to the CI helix of a neighboring PomA subunit. In four PomA subunits, we could trace the entire CH motifs from residue K246 to its C-terminal end D253, with the contact between the CH and CI mainly mediated by electrostatic and hydrophobic interactions (Fig. 5g). The remaining CH motif is disordered, without any featured density observed (Fig. 5e). This disordered CH motif likely stems from the asymmetry of the PomAB assembly, where there are two PomA subunits on one side of PomB plug motifs and three on the other side and there is less space for this CH motif to interact with the neighboring PomA CI helix. The detachment of CH from CI at one intersubunit site results in the cytoplasmic domains of PomA forming an irregular pentagon, as shown by measuring distances of those charge residues responsible for FliG torque helix binding (the center of mass of K89 and R88 to the center of mass of D114 and E96) (Fig. 5f). The PomA C-terminal region is less conserved in length and sequence among stator unit subtypes (Supplementary Fig. 1b). We made a PomA C-terminal end truncation and found that PomA CH motif truncation completely abolished motility (Fig. 5h). Based on these findings and our structural analysis, we confirm that the PomA CH motif and the CH-CI interaction are critical to sustain stator unit function.

## Discussion

Since it was first revealed in *Vibrio* species that their polar flagellar motors are driven by sodium-motive force[19,58], the Na$^+$-driven stator unit has been under intense functional and structural investigations for decades[59]. While the stator unit from *V. alginolyticus* (*Va*PomAB) has served as a prototype for the Na$^+$-driven stator unit superfamily, it has so far proven refractory to structural analysis. We here determined a high-resolution structure of the Na$^+$-driven stator unit to fill this knowledge gap.

The trans conformation of the plug motifs seems to be a universal feature among the stator unit family and their structural configuration explains how their organization tightly restrain the rotation of the stator unit (Supplementary Fig. 13). The plug motifs also prevent ion influx into the cytoplasmic domain before the stator unit incorporates into the rotor. Their distinct interaction environments caused by the imbalanced PomA$_5$:PomB$_2$ subunit stoichiometry also suggest their asymmetric release during the stator-rotor interaction. The signal that promotes the periplasmic plug motif release is probably triggered by the cytoplasmic stator unit-rotor interaction upon the incorporation of the stator units into the motor, with the signal transmission route likely being through PomA transmembrane peripheral helices, particularly those two dynamic PomA TM1 helices. Plug motif release could then facilitate PomB PGB motifs dimerization, which can reach and anchor to the cell wall through recognition of the peptidoglycan components by the dimerized PGB interfacial grove, and this will produce a spatial tension preventing rebinding of the released plug motif to the activated stator unit. Therefore, only the rotor-incorporated unplugged stator units represent their fully activated states. Indeed, we were unable to purify the unplugged PomAB after deleting the PomB plug motif. Likely, the plug deletion PomAB complex did not assemble well

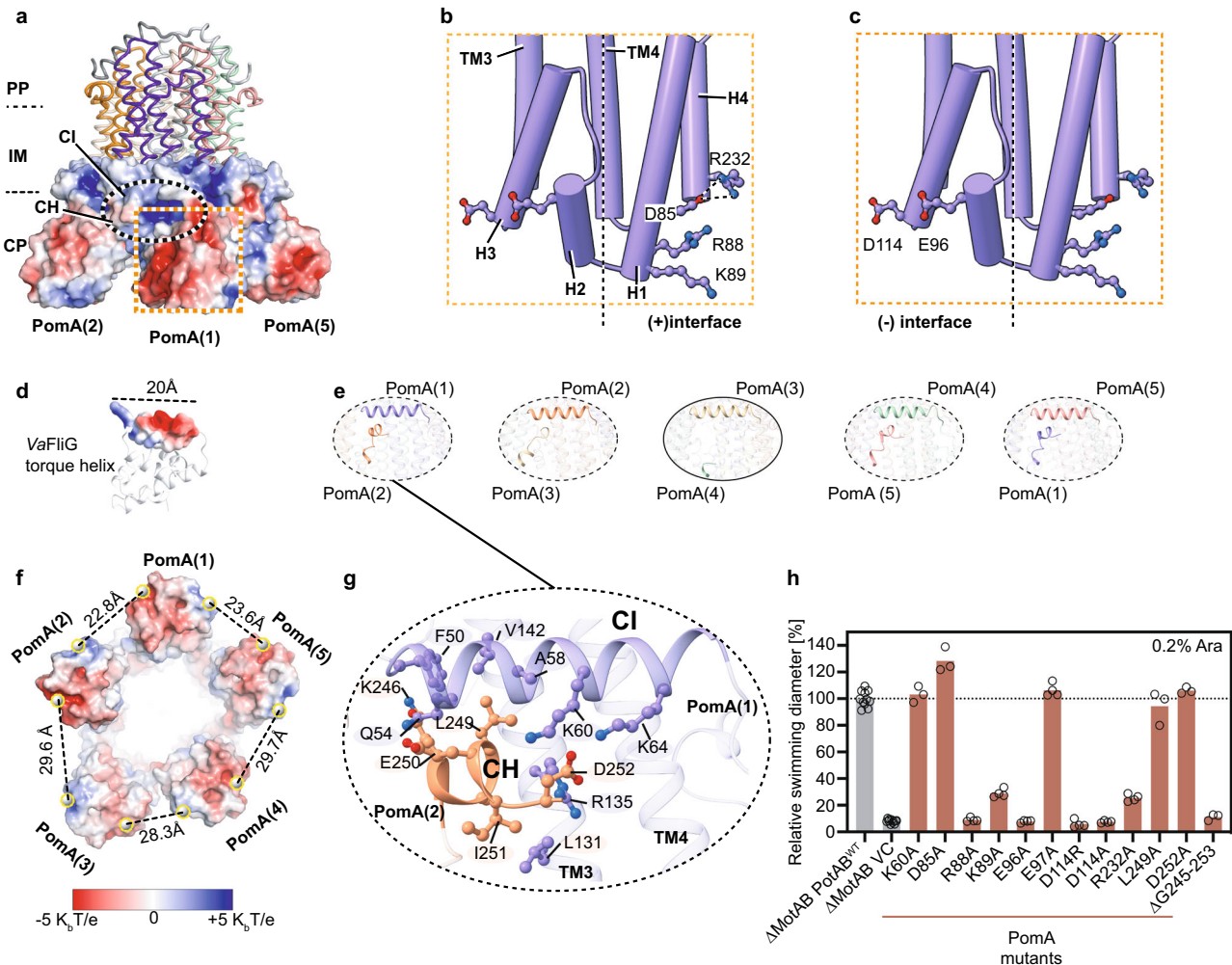

**Fig. 5 | PomA cytoplasmic domain and C-terminal helical motif. a** PomAB cytoplasmic domain electrostatic potential. **b** Locations of key residues responsible for FliG torque helix binding, highlighting the positively charged residues from the principal interface. **c**, Similar to **b**, but highlighting the negatively charged residues from the complementary interface. **d** *Va*FliG C-terminal domain (based on homology modeling) containing the torque-generating helix is shown, and its length is indicated. **e** Interactions between PomA CH helix and CI helix. One site without interaction is highlighted and circled with a solid line. **f** Image from

**a** viewed from the cytoplasmic domain. Distances between the center of mass of the residues K89, R88 and the center of mass of the residues D114, E96 from adjacent PomA subunits are given. **g** Detailed interactions between CH motif and CI helix. Residues involved in interactions are shown as sticks. **h** Comparison of motility ability of the *Va*PotAB constructs and point mutants of the residues involved in FliG torque helix interaction or PomA C-terminal truncation. Source data are provided as a Source Data file.

and was toxic to the cells due to ion leakage, and the unplugged PomAB is more stable upon rotor incorporation.

The ion permeation pathway identified in the PomAB structure provides an energy advantage by shortening the sodium ion translocation path from the periplasmic side to the key ion-accepting residue PomB D24. PomB S27, a polar residue right above D24, may increase solvent accessibility (Fig. 3d). In addition, the hydrophobic residues found at the periplasmic assembly interface of PomA and PomB may block the ion from flowing back to the periplasmic space, and they may also stabilize the stator unit by preventing it from falling apart during the stator unit's dwell-on the rotor (Fig. 4a). A recent study showed that when *E. coli* MotAB is replaced with an engineered PomAB (PomB PGB replaced with *E. coli* MotB PGB), at a low Na$^+$ environment, the engineered PomAB can rapidly aquire mutations, which restore bacterial motility[60] and reflect the adaptability of the stator unit. This is consistent with our results, where those mutations in the *Va*PotAB (PomB PGB replaced with *S. enterica* MotB PGB) granted the stator unit a gain-of-function phenotype in *S. enterica* (Supplementary Fig. 14a–b). Most of those mutation sites reside near the ion selectivity cavity (Supplementary Fig. 14c–d), including PomB G20, L28 and PomA L183, and

upon mutation may modulate the ion specificity, probably enabling the stator unit to use both Na$^+$ and H$^+$ as coupling ions. Of note, in the H$^+$-driven stator unit *C. jejuni* MotAB, the equivalent site of PomA L183 is phenylalanine (*Cj*MotA186), whose side chain adopts two conformations in the activated stator unit, affecting H$^+$ translocation efficiency[33]. We also noticed that PomB L36Q has a gain-of-function phenotype. In the plugged PomAB structure, PomB chain1 L36 hydrophobically interacts with PomB chain 2 plug motif F47 (not PomB chain2 L36 with PomB chain 1 F47, due to asymmetric assembly) (Supplementary Fig. 12c–d). The L36Q mutation possibly decreases the plug motif binding energy and makes the stator unit more activatable. In addition, it is unlikely that PomB L36 lines the previously proposed ion translocation pathway in which it forms the dehydration gate with nearby hydrophobic residues[61,62], as the L36A mutant has the same motility as the wildtype phenotype (Supplementary Fig. 14a).

The observed CH-CI interactions and the detachment in one site as well as the irregular pentagonal shape of PomA cytoplasmic domain likely contribute to the process of stator unit assembly onto the rotor. We propose the following model for the dynamic stator unit binding to the rotor, in which the stator unit randomly orients towards the rotor

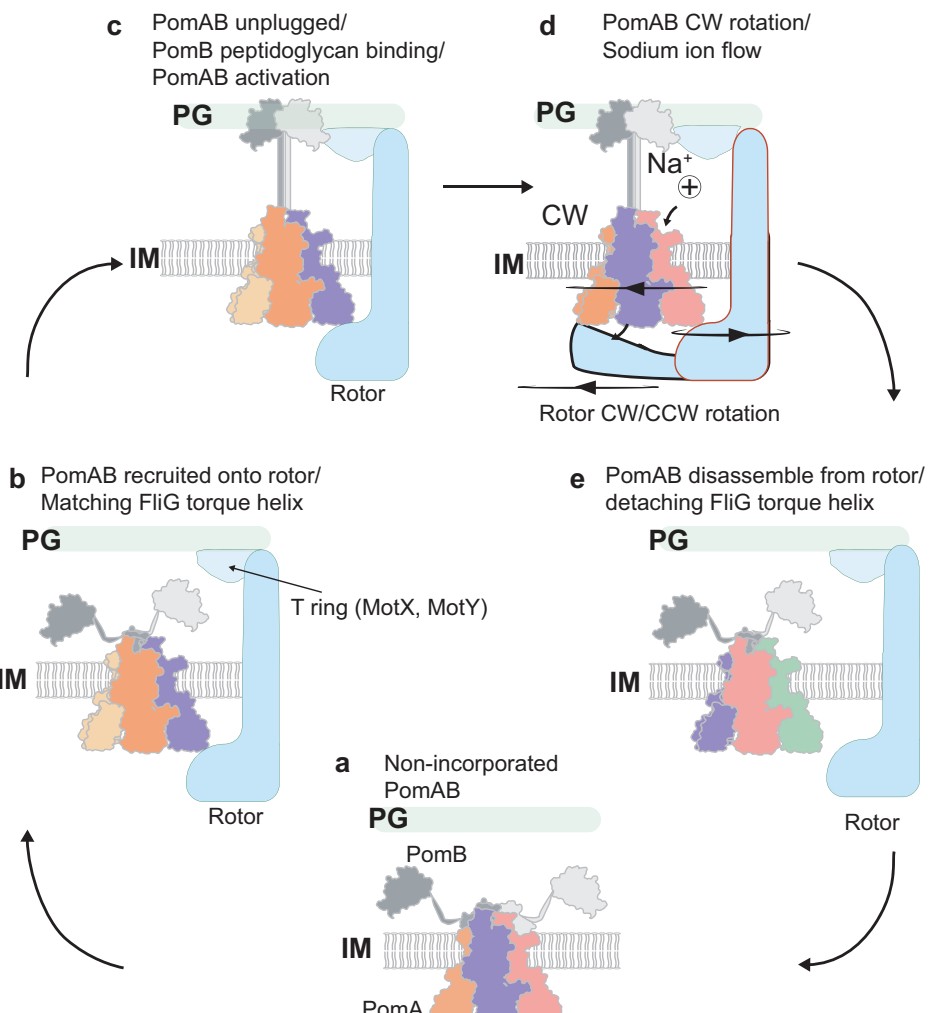

**Fig. 6 | Models of *Va*PomAB activation and disassembly from rotor. a** An inactive stator unit is plugged autoinhibited. **b** Inactive stator unit orients its cytoplasmic domain towards the rotor to contact FliG torque helix. **c** The signal from the interaction between stator unit and rotor is transferred to the PomAB periplasmic domain, where it promotes the plug motifs release, followed by PomB PGB motifs dimerization and binding to the peptidoglycan layer. PomAB gets activated.

**d** In the activated PomAB, a sodium ion (represented by a sphere with a + symbol) passes through the PomA selectivity bind filter, and binds to PomB Asp24, triggering CW rotation of PomA around PomB. The rotor could rotate either CW or CCW direction, depending on how it interacts with the stator unit. **e** Stator unit disassembly from the rotor when external torque is decreased.

and 'measures' the length of the FliG torque helix. Once both principal and complementary faces of the PomA cytoplasmic domain catch the FliG torque helix, possibly through (one of) the two shortest sites among five, which fit the length of FliG torque helix best (Fig. 5d, f), the stator unit is incorporated and is activated (Fig. 6a–c). This process could be assisted by FliL, a membrane protein recently shown to enhance the stator-rotor incorporation and stabilize the stator unit in its activated form[63,64]. During the activation, each PomA subunit near the disengaged PomB D24 supplies a Na$^+$ from the ion selectivity cavity to couple the disengaged D24 with a Na$^+$. Meanwhile, the engaged PomB D24 releases the coupled Na$^+$ and together with M155 ensures CW rotation of PomA around PomB as viewed from outside of the membrane. At the same time, PomA cytoplasmic domain progressively interact with the FliG torque helix. The rotor will either be in CCW or CW rotation mode, depending on the conformation of the C-ring (Fig. 6d).

Given the fact that stator units constantly assemble and disassemble around the rotor, depending on the requirement of external load, the asymmetric PomA cytoplasmic domain could also be advantageous for the deactivated stator unit to detach from the rotor. When the external load is decreased, which likely promotes the PGB

motif of the PomB to disconnect from the cell wall, the plug motifs of the stator unit rebind to their inhibitory sites. This signal will transfer to the stator unit cytoplasmic domain, leading to its asymmetry and weakening the interactions between the stator unit and rotor, promoting the stator unit to separate from the rotor (Fig. 6e). The proposed model is reminiscent of the recently proposed 'catch-bond' mechanism, in which the interaction/bond becomes weaker under reduced force and is enhanced by rotation of the rotor[65,66]. However, the atomic structure of the whole flagellar motor with the assembled stator units is needed to fully understand the stator unit rotor incorporation mechanism and whether the asymmetric PomA cytoplasmic domain becomes symmetric during activation remains to be further investigated (Supplementary Fig. 15a–b).

In summary, we present the structures of *Va*PomAB in both detergent and lipidic environments. The cryo-EM maps not only provide a detailed structure assembly of the Na$^+$-driven stator unit, but also enable us to assign the ion binding sites, which in turn allows us to address the enigmatic mechanism of stator unit ion selectivity. Our structural analysis and functional experiments support that the stator unit is a CW unidirectional rotary motor and this is achieved by a hydrophobic directional rotation 'reinforcement' point. The PomB

plug motifs organization and discovery of PomA C-terminal helical motif further expand our view about the stator unit activation and rotor incorporation.

## Methods

### *Va*PomAB purification with LMNG detergent

The DNA sequence coding for *Va*PomAB was amplified from *Vibrio alginolyticus* (ATCC 17749) and subcloned into a modified pET vector containing a C-terminal twin-Strep-tag. A human rhinovirus (HRV) 3C protease cleavage site (GTLEVLFQGPGGS) was inserted between the PomB plug motif and the peptidoglycan binding domain (between residues Gln95 and Gln96). PomAB complex was expressed in *E. coli* Overexpress™ C43(DE3) cells (LuBioScience GmbH). Cells were cultured in 8 l LB medium supplemented with 50 µg/ml ampicillin at 37 °C, and protein expression was induced with 0.5 mM IPTG at $OD_{600}$ 0.6. Cells were incubated for another 16 h at 20 °C before harvesting. The cell pellet was resuspended in buffer A (20 mM HEPES pH 7.5, 300 mM NaCl) with 30 µg/ml of DNase I and 50 µg/ml of lysozyme and incubated at 4 °C for 30 min before passaging it through an EmulsiFlex-C5 homogenizer at 15,000–20,000 pound-force per square inch. Membranes were then sedimented at $18,000 \times g$ for 1 h and stored at −20 °C after flash freezing with liquid nitrogen.

For protein purification, membranes were solubilized in buffer A supplemented with 2% (w/v) Lauryl Maltose Neopentyl Glycol (LMNG), 10% glycerol, and protease inhibitors (protease inhibitor cocktail tablets, EDTA-free, Roche Diagnostics GmbH) for 2 h at 4 °C while shaking on a rocking platform, and then ultracentrifuged for 30 min at $18,000 \times g$. The supernatant was added to a gravity flow column containing 2 ml Strep-Tactin® Superflow® resin (IBA) pre-equilibrated with washing buffer (buffer A with 10% glycerol and 0.005% LMNG). Resins were washed five times with 4 column volumes of washing buffer and Strep tagged protein was eluted with elution buffer (Buffer A, 10% glycerol, 0.005% LMNG and 10 mM desthiobiotin). The protein complex was then concentrated until reaching a volume of 0.5 ml. HRV-3C protease was added to the *Va*PomAB sample, with a protein:protease ratio of 5:1 (w/w) and incubated at 4 °C overnight. The sample was loaded onto a Superose® 6 Increase 10/300 GL (Merck) column, pre-equilibrated with buffer A with 0.002% LMNG. The peak fractions corresponding to the protein complex were concentrated to about 16–20 mg/ml using a centrifugal filter with a PES membrane (Sartorius) and used for preparation of cryo-EM sample grids immediately.

### *Va*PomAB MSP1D1 and saposin lipid nanodisc reconstitution

To reconstitute *Va*PomAB into lipid nanodiscs with MSP1D1, 500 µl of 2 mg/ml purified *Va*PomAB without PomB PGB was mixed with *E. coli* polar lipids and MSP1D1 in a molar ratio of 1:156:6.25 (*Va*PomAB:lipids:MSP1D1). The reaction was incubated at 4 °C with mild agitation for 5 min. Bio-beads (300 mg per ml reaction) were added and incubated overnight to remove the detergent. Bio-beads were filtered out the next day using a PVDF 0.22 µm Centrifugal Filter (Durapore) tube. The sample was then injected into a Superose® 6 Increase 10/300 GL (Merck) column, which was pre-equilibrated with buffer A. The peak fractions corresponding to the protein complex in lipid nanodiscs of MSP1D1 were pooled, concentrated and used for cryo-EM grids preparation.

To reconstitute *Va*PomAB into lipid nanodiscs with saposin, 300 µl of 6 mg/ml full length purified *Va*PomAB (without protease insertion) was mixed with *E. coli* polar lipids (10 mM; 200 µl) and incubated at room temperature for 10 min. Saposin (6.7 mg/ml; 350 µl) was added into the reaction and incubated for 2 min. The molar ratio of PomAB, lipids and saposin was 1:300:35, respectively. The reaction was diluted with 2 ml buffer A to initiate the reconstitution and incubated on ice for an additional 30 min. 700 mg of bio-beads were added and incubated overnight to remove the detergent. The rest of the steps were the same as when *Va*PomAB was reconstituted into MSP1D1 nanodiscs.

### Cryo-EM grids preparation and cryo-EM data collection

To break the preferential particle orientation, 0.0125% CHAPSO (final concentration) was added into the sample before grid preparation. 2.7 µl of freshly purified sample was applied onto glow-discharged (30 s, 5 mA) grids (Quantifoil R 0.6/1 300 mesh Cu or Ultrafoil 0.6/1 300 mesh Au) and plunge-frozen into liquid ethane using a Vitrobot Mark IV (FEI, Thermo Fisher Scientific) with the following parameters: 4 °C, 100% humidity, 7 s wait time, 4–4.5 s blot time, and a blot force of 25. Movies were collected using the semi-automated acquisition program EPU (FEI, Thermo Fisher Scientific) on a Titan Krios G2 microscope operated at 300 keV paired with a Falcon 3EC direct electron detector (FEI, Thermo Fisher Scientific). Images were recorded in an electron counting mode, at 96,000x magnification with a calibrated pixel size of 0.832 Å and defocus range of 0.8–3 µM. For the *Va*PomAB sample purified in LMNG, 6467 micrographs were collected, with each micrograph containing 40 frames and a total exposure dose of 37.98 (e/Å$^2$). For the *Va*PomAB sample reconstituted into saposin nanodiscs, 3927 micrographs were collected, with each micrograph containing 40 frames and a total exposure dose of 37 (e/Å$^2$). For the *Va*PomAB MSP1D1 sample, 5450 micrographs were collected, with each micrograph containing 40 frames and a total exposure dose of 40 (e/Å$^2$).

### Image processing

To keep the image data processing consistent, all the datasets were processed using cryoSPARC version 3.3.2 and version 4, unless otherwise stated. Patch motion correction was used to estimate and correct frame motion and sample deformation (local motion). Patch Contrast function (CTF) estimation was used to fit local CTF to micrographs. Micrographs were manually curated to remove the bad ones (relatively ice thickness thicker than 1.05 and CTF value worse than 3.2 Å for LMNG dataset; relatively ice thickness thicker than 1.1 and CTF value worse than 5 Å for MSP1D1 nanodisc dataset; relatively ice thickness thicker than 1.2 and CTF value worse than 5 Å for Saposin nanodisc dataset). Particles were picked using the Topaz software implemented in cryoSPARC[67]. Basically, Topaz extract was used with a pre-trained model with a pre-tested particle threshold value. Particles were extracted with a box size of 400 pixels and Fourier crop to box size of 100 pixels. Duplicated particles were removed using a minimum separation distance criteria of 60 Å, which means that the distance between the centers of two neighboring particles should be larger than 60 Å. One round of 2D classification was then performed, followed by ab-initio reconstruction. Heterogeneous refinement was used to get rid of the junk particles. Particles were re-extracted with full box size (400 pixels). Non-uniform refinement was applied with a dynamic mask to obtain a high-resolution map. Local refinement was additionally performed with a soft mask surrounding *Va*PomAB complex in order to achieve a higher resolution map. The number of micrographs, total exposure values, number of particles used for final refinement, and map resolution values for all datasets are summarized in Table S1.

### Atomic model building, refinement, and validation

ColabFold[68] was used to predict the structure of PomA pentamer[69] and manually fit the model into the density by using UCSF ChimeraX[70]. The model was refined in Coot[71], and PomB TM and plug motif was manually modeled. The model was then refined against the map using PHENIX real space refinement[72].

### Molecular dynamics simulation of PomAB

The system was constructed by embedding the cryo-EM structure of PomAB into a flat, mixed lipid bilayer consisting of 16:0-18:1 phosphatidylethanolamine (POPE) and 1-palmitoyl-2-oleoyl

phosphatidylglycerol (POPG) at a 4:1 ratio using the Membrane Builder tool of CHARMM-GUI webserver[73]. Explicit water was added using the TIP3P water model, and the system charge was neutralized with sodium ions and solvated in a cubic water box containing 0.15 M NaCl. The size of the box was 11.0, 11.0, and 11.5 nm in the x, y and z dimension, respectively, resulting in ~144,000 atoms in total. The CHARMM36m force field[74] was used for the protein, and the CHARMM36 lipid force field[75] was used for all lipid molecules. Note that the WYF correction was included in the force field to improve the description of the cation-π interactions[76]. The temperature was kept constant at 310 K using the V-rescale algorithm with a 2 ps coupling constant, and the pressure at 1.0 bar using the Parrinello-Rahman barostat[77] with a 5 ps time coupling constant. A cutoff of 1.2 nm was applied for the van der Waals interactions using a switch function starting at 1.0 nm. The cutoff for the short-range electrostatic interactions was also at 1.2 nm and the long-range electrostatic interactions were calculated by means of the particle mesh Ewald decomposition algorithm with a 0.12 nm mesh spacing. A reciprocal grid of $96 \times 96 \times 96$ cells was used with 4th order B-spline interpolation. MD simulations were performed using Gromacs2021.5[78]. Two independent simulations were performed, each for one μs. Analysis of the MD trajectories was performed using the Gromacs gmx and GROmaρs tools[79].

The relative free energy difference between Na$^+$ and K$^+$ in the sodium binding cavity (the right binding site in Fig. 4d) was calculated using the free energy perturbation method[80]. A hybrid topology that contains both ions was prepared for the alchemical transformations of the atom types. To simplify the calculation, a 'double-system/single-box' setup[81] was used, so that $\Delta\Delta G_{binding}^{N\rightarrow K+}$ could be obtained in a single set of calculations. The alchemical transitions were performed with thirteen lambda windows and each window was run for 4 ns. The calculation was repreated 3 times to confirm consistency.Gromacs 2021.5 was used to carry out the alchemical simulations and *gmx bar* tool was used to analyze them.

## Bacterial strains and growth

*Escherichia coli* and *Salmonella enterica* serovar *Typhimurium* LT2 (J. Roth) (ATCC 700720) were grown at 37 °C with aeration at 180 rpm in lysogeny broth (LB medium) [10 g/l tryptone, 5 g/l yeast extract and 5 g/l NaCl]. For solid agar plates, 1.5% (w/v) of agar-agar was added, alternatively to test swimming motility 0.3% (w/v) of agar-agar was supplemented. All strains used in this study are listed in the supplement information Table S2. For strains harboring a plasmid carrying a resistance marker selected media were supplemented with chloramphenicol (12.5 μg/ml). Induction experiments were performed in the presence of arabinose (0.2%).

## DNA manipulation

Plasmids were constructed according to standard cloning techniques as described elsewhere (ISBN 0879695773). In brief, rolling circle, around the horn PCR and overlap PCR were applied to generate point mutations in *pomA* or *pomB*, respectively. The primers used in this study are listed in the supplement information Table S3. For DNA amplification Q5 polymerase was used and for verification OneTaq polymerase (both purchased from NEB, Ipswich, MA, USA). All plasmids were verified by sequencing.

## Motility assay

To assess the swimming motility of *Va*PotAB mutants, strains were inoculated in liquid LB medium supplemented with chloramphenicol and grown overnight. 2 μl of the overnight cultures were inoculate into soft agar plates containing the selective marker and supplemented with or without arabinose and incubated at 37 °C. Once a motility halo was visible, plates were scanned. From these images, swimming diameters were measured using Fiji[82].

## Figure preparation

Figures were prepared using ChimeraX 1.6.1[70], PyMOL 2.5.4, GraphPad Prisim 9.4.1 and Adobe Illustrator. Surface buried area and solvation free energy was calculated using the online webserver PDBePISA[83]. Structural comparison between PomAB-LMNG and PomAB-MSP1D1 models was performed in Pymol by aligning 1312 C$_\alpha$ atoms, without outlier rejection. The electrostatic potential maps were calculated using the APBS[84] electrostatic Plugin integrated inside Pymol. Briefly, *Va*PomAB high resolution model and *Va*FliG AlphaFold model were prepared using "pdb2pqr"[85] by assigning partial charged and hydrogens as well as missing atoms. Then the electrostatic maps were calculated using APBS with grid spacing 0.5. The calculated map was projected onto the corresponding molecule with the scale given in unit $K_bT/e$, where $K_b$ represents Boltzmann's constant, T is temperature in Kelvin and e is the charge of an electron. We have provided in Supplementary Fig. 16. The uncropped and unprocessed SDS gels presented in Supplementary Fig. 2b, Supplementary Figs. 3a and 4a.

## Reporting summary

Further information on research design is available in the Nature Portfolio Reporting Summary linked to this article.

## Data availability

Atomic coordinates for *Va*PomAB in LMNG detergent and *Va*PomAB in MSP1D1 nanodisc were deposited in the Protein Data Bank under accession codes PDB: 8BRD and 8BRI, respectively. The corresponding electrostatic potential maps were deposited in the Electron Microscopy Data Bank (EMDB) under accession codes EMDB: EMD-16212 and EMD-16215, respectively. The electrostatic potential map for full-length *Va*PomAB in Saposin nanodisc was deposited in the EMDB under accession code EMDB: EMD-16214. Source data are provided with this paper.

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

## Acknowledgements

The Novo Nordisk Foundation Center for Protein Research is supported financially by the Novo Nordisk Foundation (NNF14CC0001). N.M.I.T. acknowledges support from DFF grant (8123-00002B) and NNF Hallas-Møller Emerging Investigator grant (NNF17OC0031006). M.E. acknowledges support by the European Research Council (ERC) under the European Union's Horizon 2020 research and innovation program (agreement no. 864971). Y.W. acknowledges the financial support from National Key Research and Development Program of China (No. 2021YFF1200404) and the Fundamental Research Funds for the Central Universities of China (No. K20220228), as well as the access to computational resources from the Information Technology Center and State Key Lab of CAD&CG, ZheJiang University. H.H. acknowledges support from Lundbeck Foundation postdoc R347-2020-2429. We thank the Danish Cryo-EM Facility at the Core Facility for Integrated Microscopy (CFIM) at the University of Copenhagen and Tillmann Pape and Nicholas Heelund Sofos for support during data collection.

## Author contributions

N.M.I.T. and H.H. conceived the project and designed experiments. H.H. expressed, purified, optimized, prepared cryo-EM grids, collected cryo-EM data and determined the structure of *Va*PomAB and the structures of *Va*PomAB in nanodiscs. M.S. helped with protein expression, purification, and cryo-EM grid preparation at the beginning of this project. P.F.P. performed the motility assay and together with M.E. interpreted data. Y.W. and Z.L. performed the molecular dynamics simulations. M.S., A.R.-E., Y.M.Y., and F.J.O.M. helped with data analysis and figure preparation. N.W. helped with data interpretation. H.H. built and refined the structure models, prepared figures and wrote the first draft of the manuscript with input from all the authors, which was then edited by N.M.I.T. and M.E. All authors contributed to the revision of the manuscript.

## Competing interests

The authors declare no competing interests.
