## [Peer Review File · Nature Communications]

Ion selectivity and rotor coupling of the Vibrio flagellar sodium-driven stator unitREVIEWER COMMENTS

Reviewer #1 (Remarks to the Author):

Please see my comments in the attached MS Word file.

Reviewer #2 (Remarks to the Author):

In this manuscript Hu et al. determine cryo-EM structures of sodium-driven stator complex (PomAB) in *Vibrio* at ~2Å resolution and reveal sodium ion binding sites, providing new insights into the ion selectivity of the stator complex. Importantly, the authors provide evidence suggesting that the bulky hydrophobic side chain of PomA M155 has an important role in keeping the stator units only in clockwise rotation. This is an excellent study that not only represents a major advance in understanding the structure and function of the sodium-driven stator complex in *Vibrio*, but also reveals some general principles underlying the stator complexes in different bacterial species. The manuscript is well-written, and structures and figures are excellent. The work will be likely appreciated by a broad audience of microbiologists. Below are some specific comments for consideration.

Line 86: “overall architecture of VaPomAB” does not accurately represent the cryo-EM structure as the major portion of PomB in periplasmic space was removed.

Line 87: “Intact VaPomAB is an anisotropically shaped complex and shows preferential orientation of particles in vitreous ice.” The data were not presented in the main figures or supplemental figures.

Fig. 1: Cryo-EM structure at 2Å resolution should be further validated by comparing zoom-in cryo-EM density maps and atomic models, especially at the ion-binding sites. The topology diagram in panel f is not scaled appropriately.

Similarly, it's important to show the densities that are formed by the water molecules or ions.

It might be beneficial to thoroughly compare the structures of the sodium-driven stator complex and the proton-driven stator complexes.

Remarks to the authors

This manuscript by Hu et al. presents the structure of the *Vibrio* flagellar stator unit PomAB at a resolution of ~2.5 Å using single-particle cryo-electron microscopy. This structure revealed a 5(PomA):2(PomB) stoichiometry and a self-inhibited plugged state of the PomAB complex, both of which are universal among the stator unit family. The structure also revealed nonresidue densities, which the authors modeled as Na⁺ ions. By combining structural and electrostatic potential analyses, all-atom molecular dynamics simulations, and mutagenesis, the authors identified the key residues for ion selectivity and proposed a Na⁺ translocation pathway. In addition, the authors performed structural and functional analyses to show that a bulky hydrophobic side chain from PomA is required for the bacterial motility, presumably by sterically blocking the CCW rotation and ensuring a unidirectional CW rotation. Finally, the asymmetric pentagonal shape of the PomA cytoplasmic domains allowed the authors to propose a model of how the stator units dynamically bind to and detach from the rotors. In summary, this manuscript is a solid piece of work that will be of broad interest to the readers of Nature Communications.

I have the following suggestions/questions to the authors:

1. Lines 52–53 (...stator units use potassium ... have also been reported): please change “use” to “using” to make the sentence grammatically correct.
2. Line 57 (...functional views of stator unit assemble, torque generation, and ...): please use the word “assembly” instead of “assemble” since the sentence needs a noun phrase here.
3. Line 64 (e.g. *V. cholerae*, *V. alginolyticus*): please add a comma (“,”) after “e.g.” to be consistent with the usage of “e.g.,” in line 52.
4. Line 65–66 (... has been linked to their virulence and biofilm formation.): I’m not requesting this, but the authors may want to consider adding “and dispersal” to the end of this sentence and referencing relevant works, since biofilm dispersal also requires motility and is a key step of the biofilm lifecycle.
5. Line 80 (a helical motif C-terminal of PomA): I suggest changing it to “a helical motif in the C-terminal region of PomA.”
6. Line 88–90 (To improve sample homogeneity, ... during protein purification.): This is the second sentence of the Results section, and it’s already talking about the plug region and the PGB domain of PomB. I would suggest adding a reference to Fig. S1 and/or Fig. 1 to help the readers (especially people who’re not familiar with this field) understand what these two regions are.
7. Line 93: The authors need to explain briefly what the LMNG detergent is.
8. Line 95 (...approaches to 2 Å): please remove the word “to”.
9. Line 105 (...TM1 of one PomA makes prominent contact with the TM2 from the adjacent subunit.): Since it’s still early in the manuscript, I would really appreciate if the authors could add an illustration/annotation of this contact in Fig. 1b to help the readers understand the spatial organization of the complex.

10. The coloring of the PomA subunits in Fig. S5 (purple=1, orange=5, yellow=4, green=3, and red=2) is not consistent with that in the main figures (purple=1, orange=2, yellow=3, green=4, and red=5). I suggest making the coloring consistent to help the readers going back and forth between the main text and the SI.
11. Line 165 (root-mean square deviation): how are the two structures aligned and how is the RMSD calculated (especially when the resolutions of both structures are larger than the RMSD)? The authors need to provide a bit more details in the Methods section about this.
12. Line 190 (...near PomA A190): I suggest annotating A190 in Fig. 3b.
13. Regarding Figs. S6 and S7, I have two suggestions/questions:
 - a. I suggest adding a brief description (in words or cartoons) what the two angles χ_1 and χ_2 represent. This will help readers from outside the field better understand the characterization.
 - b. Why does Fig. S6 only show χ_1 and Fig. S7 only show χ_2 ? Does it make sense to present the simulation trajectories on χ_1 - χ_2 plots?
14. In Figs. 3–5, I'm assuming that " Δ MotAB VC" denotes the negative control with empty vectors, but the authors need to clarify what it means.
15. Lines 235–237 (...substituting any of these three threonines to alanines abolishes bacterial motility,...): I suggest referring to Fig. 14a about the result of the T158A mutant, which is currently not in Fig. 3i.
16. Lines 237–243: These arguments for why the Na⁺ cavity has ion selectivity make sense. But I'm wondering if the authors could use the MD simulations to support their arguments (again, I'm not requesting this, just thinking about the possibility.)
17. Lines 247–248: Would it be possible to compare the Na⁺ binding site 2 in *V. alginolyticus* with the corresponding sites in *C. jejuni* MotAB and *B. subtilis* MotAB, in a manner similar to that shown in Fig. 3f–h? If so, the authors might want to consider adding this comparison to the SI.
18. Lines 271–275: Have the authors tried to mutate the G residues in the glycine zipper and see how that affects the bacterial motility?
19. Lines 353–354 (Besides, R232 establishes an interdomain salt bridge with residue D85, ...): Have the authors tried to mutate D85 and see how this mutation affects the bacterial motility?
20. I couldn't find how the authors computed the electronic potential maps. The authors need to provide details in the Methods section about this.

Point-by-point response to reviews' comments

"Ion selectivity and rotor coupling of the *Vibrio* flagellar sodium-driven stator unit" (Hu *et al.*) - NCOMMS-23-02151

We would like to thank the two reviewers for their careful assessment of our manuscript and for their insightful comments that have been very helpful to improve the quality of this research article. We here present a response to all the points raised by the reviewers.

Reviewer #1 (remarks to the author):

This manuscript by Hu et al. presents the structure of the Vibrio flagellar stator unit PomAB at a resolution of ~2.5 Å using single-particle cryo-electron microscopy. This structure revealed a 5(PomA):2(PomB) stoichiometry and a self-inhibited plugged state of the PomAB complex, both of which are universal among the stator unit family. The structure also revealed nonresidue densities, which the authors modeled as Na⁺ ions. By combining structural and electrostatic potential analyses, all-atom molecular dynamics simulations, and mutagenesis, the authors identified the key residues for ion selectivity and proposed a Na⁺ translocation pathway. In addition, the authors performed structural and functional analyses to show that a bulky hydrophobic side chain from PomA is required for the bacterial motility, presumably by sterically blocking the CCW rotation and ensuring a unidirectional CW rotation. Finally, the asymmetric pentagonal shape of the PomA cytoplasmic domains allowed the authors to propose a model of how the stator units dynamically bind to and detach from the rotors. In summary, this manuscript is a solid piece of work that will be of broad interest to the readers of Nature Communications. I have the following suggestions/questions to the authors:

We thank the reviewer for their positive feedback and detailed assessment.

1. *Lines 52–53 (...stator units use potassium ... have also been reported): please change “use” to “using” to make the sentence grammatically correct.*

We have made this correction.

2. *Line 57 (...functional views of stator unit assemble, torque generation, and ...): please use the word “assembly” instead of “assemble” since the sentence needs a noun phrase here.*

We have corrected this.

3. *Line 64 (e.g. *V. cholerae*, *V. alginolyticus*): please add a comma (“,”) after “e.g.” to be consistent with the usage of “e.g.,” in line 52.*

Done

4. *Line 65–66 (... has been linked to their virulence and biofilm formation.): I’m not requesting this, but the authors may want to consider adding “and dispersal” to the end of this sentence*

and referencing relevant works, since biofilm dispersal also requires motility and is a key step of the biofilm lifecycle.

We agree with the reviewer that motility of the flagellated bacteria plays a crucial role not only in their biofilm formation, but also in biofilm dispersion. We have changed the sentence to “*has been linked to their virulence, biofilm formation and dispersion*”. At the end of this sentence, we have added a citation (now reference 42): ‘Mechanisms Underlying *Vibrio cholerae* Biofilm Formation and Dispersion’.

5. *Line 80 (a helical motif C-terminal of PomA): I suggest changing it to “a helical motif in the C- terminal region of PomA.”*

We have rephrased the sentence and changed it to “*a helical motif in the C- terminal region of PomA.*”.

6. *Line 88–90 (To improve sample homogeneity, ... during protein purification.): This is the second sentence of the Results section, and it’s already talking about the plug region and the PGB domain of PomB. I would suggest adding a reference to Fig. S1 and/or Fig. 1 to help the readers (especially people who’re not familiar with this field) understand what these two regions are.*

We thank the reviewer’s suggestion; we have referenced **Fig. 1** and **Fig. S1b** at the end of the sentence to show what are the PomB plug motif and PGB domain. Additionally, in **Fig. S1b**, at the protein sequence level, we highlighted where we inserted the HRV-3C protease site.

7. *Line 93: The authors need to explain briefly what the LMNG detergent is.*

We have added “Lauryl Maltose Neopentyl Glycol” before the abbreviation LMNG to clarify this.

8. *Line 95 (...approaches to 2 Å): please remove the word “to”.*

Done.

9. *Line 105 (...TM1 of one PomA makes prominent contact with the TM2 from the adjacent subunit.): Since it’s still early in the manuscript, I would really appreciate if the authors could add an illustration/annotation of this contact in Fig. 1b to help the readers understand the spatial organization of the complex.*

We presume the reviewer is referring to **Fig. 1c**, in which we have now annotated the secondary structural elements in the context of the protein model to better present PomAB structure.

10. The coloring of the PomA subunits in Fig. S5 (purple=1, orange=5, yellow=4, green=3, and red=2) is not consistent with that in the main figures (purple=1, orange=2, yellow=3, green=4, and red=5). I suggest making the coloring consistent to help the readers going back and forth between the main text and the SI.

We thank the reviewer pointing this out. We have modified the coloring of the PomA subunits in **Fig. S5**. We also have modified the coloring in **Fig.S11** and **Fig. S13** to make the coloring consistent between the main text and the supplementary information.

11. Line 165 (root-mean square deviation): how are the two structures aligned and how is the RMSD calculated (especially when the resolutions of both structures are larger than the RMSD)? The authors need to provide a bit more details in the Methods section about this.

We apologize for lack of clarity in the Methods section about how we aligned the two structures. We have revised the text to state: root-mean-square deviation of 0.642 Å, 1,312 Ca atoms aligned.

In the Methods section we added: “*Structural comparison between PomAB-LMNG (Model_1) and PomAB-MSP1D1(Model_2) models was performed in Pymol by aligning 1,312 Ca atoms, without outlier rejection.*”

Another point raised by reviewer is how the RMSD value reflects the difference of the two models when the resolution of both structures is larger than the RMSD. This question can be addressed by calculating position-dependent Z-score, which involves using the model B-factor in the following manner:

$$Z(i) = \frac{\Delta R(i)}{\sigma(i)}$$

Where $\Delta R(i)$ is the distance between Ca atoms of residue (i) in the two forms Model_1 and Model_2:

$$\Delta R(i) = |R_i(model_1) - R_i(model_2)|$$

The error (σ) on $\Delta R(i)$ can be calculated as:

$$\sigma(i) = \sqrt{\sigma^2_i(model\ 1) + \sigma^2_i(model\ 2)}$$

Where σ^2 is the mean-square atomic displacement, measured in \AA^2 , and its relationship with B-factor (B) is:

$$\sigma^2 = 3B/8\pi^2$$

Plot of the B-factor vs residue across the two PomAB structures: LMNG (pink) and MDSP1D1 (teal). Grey dot-dashed vertical lines represent the end/beginning of each chain.

To calculate the Z-score (i) based off the Ca distance and B-factors we can rearrange the above equations to:

$$z(i) = \frac{2\pi\sqrt{2}\Delta R(i)}{\sqrt{3B_i(\text{model 1}) + 3B_i(\text{model 2})}}$$

Plot of the position-dependent Z-score for each residue. The grey dot-dashed vertical lines represent the end/beginning of each chain. The red dashed line represents the average Z-score across the two structures.

We calculated this position-dependent Z-score for each residue, then its mean value (0.358 ± 0.198) plot is shown above, and of note, a p-value of 0.1 (yellow dashed lines) and 0.05 (black dashed lines) corresponds to Z-scores of 1.65 and 1.96, respectively (plots are shown below), and all the position-dependent Z-scores fall substantially below these thresholds, indicating the difference between the PomAB LMNG and PomAB MSP1D1 models is not significant.

12. Line 190 (...near PomA A190): I suggest annotating A190 in Fig. 3b.

Thanks for the suggestion. We have annotated A190 in Fig. 3b.

13. Regarding Figs. S6 and S7, I have two suggestions/questions:

a. I suggest adding a brief description (in words or cartoons) what the two angles χ_1 and χ_2 represent. This will help readers from outside the field better understand the characterization.

We have clarified the definitions of the two torsion angles χ_1 and χ_2 in the figure legends S6 and S7: The χ_1 angle of T158, T185 and T186 is the angle between the planes formed by N-C α -C β and C α -C β -O γ atoms, while the χ_2 angle of D24 is the angle between the planes formed by C α -C β -C γ and C β -C γ -O δ atoms.

b. Why does Fig. S6 only show χ_1 and Fig. S7 only show χ_2 ? Does it make sense to present the simulation trajectories on χ_1 - χ_2 plots?

We calculated the side chain dihedral angles χ_1 and χ_2 for several key residues, depending on the length of their side chains. For Asp, both χ_1 and χ_2 are available, but χ_2 better reflects the side chain dynamics and is shown in Fig. S6. Thr residues (T158, T185 and T186) have shorter side chains than Asp residues (D24), so only χ_1 is meaningful for Thr and shown in Fig. S7.

14. In Figs. 3–5, I’m assuming that “ Δ MotAB VC” denotes the negative control with empty vectors, but the authors need to clarify what it means.

The reviewer is correct. “ Δ MotAB VC” stands for an empty vector transformed into a mutant

Salmonella enterica strain that lacks its wildtype MotAB and serves as a negative control. We have added this information in the figure legend.

15. Lines 235–237 (...substituting any of these three threonines to alanines abolishes bacterial motility,...): I suggest referring to Fig. 14a about the result of the T158A mutant, which is currently not in Fig. 3i.

We thank the reviewer for pointing this out. We have added the motility phenotype of the PomA T158A mutant in panel **Fig. 3i** and reference this figure in the appropriate place in the text.

16. Lines 237–243: These arguments for why the Na⁺ cavity has ion selectivity make sense. But I'm wondering if the authors could use the MD simulations to support their arguments (again, I'm not requesting this, just thinking about the possibility.)

We thank the reviewer for this suggestion. To support our arguments, we have performed free energy perturbation calculations to compare the binding free energies of Na⁺ and K⁺ in the sodium selectivity cavity. We found that Na⁺ has a lower binding free energy than K⁺ by 4.37±0.7 kJ/mol, indicating that this cavity prefers Na⁺ over K⁺. We have included these results in our revised manuscript.

17. Lines 247–248: Would it be possible to compare the Na⁺ binding site 2 in *V. alginolyticus* with the corresponding sites in *C. jejuni* MotAB and *B. subtilis* MotAB, in a manner similar to that shown in Fig. 3f–h? If so, the authors might want to consider adding this comparison to the SI.

This is indeed a good point. We have added in **Fig. S8g-i** structure comparisons of the Na⁺ binding site 2 in *V. alginolyticus* PomAB, with the corresponding sites in *C. jejuni* MotAB and *B. subtilis* MotAB. Clearly, in the proton-driven stator unit, the equivalent site lacks a Na⁺-contacting environment, because two threonines, T158 and T185 (*Va*PomA sequence numbering system), are replaced by alanine in *Cj*MotAB and *Bs*MotAB, which are proton-driven stator units.

18. Lines 271–275: Have the authors tried to mutate the G residues in the glycine zipper and see how that affects the bacterial motility?

This is certainly a good suggestion. Indeed, we point out that the glycine zipper GXXGXXXG (in PomA, residues G154-G161) is a universally conserved motif in the flagellar stator unit, and it lines the ion selectivity filter. We generated several point mutations of this motif (G154L, G157L, G161L) and observed each individual mutation abolished bacterial motility. Please see **Fig. S14a** and the figure below, in which the three mutants are highlighted with a red star.

19. Lines 353–354 (Besides, R232 establishes an interdomain salt bridge with residue D85, ...Have the authors tried to mutate D85 and see how this mutation affects the bacterial motility?)

Positively charged residues from one PomA subunit contribute to the principal face or (+) face, which binds negatively charged residues of the FliG torque helix. R232 and D85 are located on the (+) face and form an interdomain salt bridge, as shown by our structural analysis. We found that the R232A mutation impairs mobility. Additionally, we generated a D85A mutation and observed the D85A mutant displayed a gain-of-function phenotype (see figure below), consistent with the fact that electrostatic interactions are the main driving factor for the stator-rotor binding. Suppressing the charge of D85 likely increases the binding of the FliG torque helix. We clarified this in the text by stating that: “Besides, R232 establishes an interdomain salt bridge with residue D85, which is probably not only stabilizing helix bundle organization, but also contributing to the FliG torque helix binding. The R232A mutation impairs mobility, and the D85A mutation results in a gain-of-function phenotype (Fig. 5b).”

20. I couldn't find how the authors computed the electronic potential maps. The authors need to provide details in the Methods section about this.

We thank the reviewer for pointing this out. We include the following statement now in the Materials and Methods section and cite the relevant papers. The electrostatic potential maps were calculated using the APBS electrostatic Plugin integrated inside Pymol. Briefly, *Va*PomAB high resolution model and *Va*FliG alphafold model were prepared using “pdb2pqr” by assigning partial charged and hydrogens as well as missing atoms. Then the electrostatic maps were calculated using APBS with grid spacing 0.5. The calculated map was projected onto the corresponding molecule with the scale

given in unit $k_b T/e$, where k_b represents Boltzmann's constant, T is temperature in Kelvin and e is the charge of an electron.

Reviewer #2 (Remarks to the Author):

In this manuscript Hu et al. determine cryo-EM structures of sodium-driven stator complex (PomAB) in Vibrio at ~2Å resolution and reveal sodium ion binding sites, providing new insights into the ion selectivity of the stator complex. Importantly, the authors provide evidence suggesting that the bulky hydrophobic side chain of PomA M155 has an important role in keeping the stator units only in clockwise rotation. This is an excellent study that not only represents a major advance in understanding the structure and function of the sodium-driven stator complex in Vibrio, but also reveals some general principles underlying the stator complexes in different bacterial species. The manuscript is well-written, and structures and figures are excellent. The work will be likely appreciated by a broad audience of microbiologists. Below are some specific comments for consideration. suggestions/questions to the authors:

We thank the reviewer for their positive feedback, detailed assessment, and constructive critique.

- 1. Line 86: “overall architecture of VaPomAB” does not accurately represent the cryo-EM structure as the major portion of PomB in periplasmic space was removed.*

We have revised this sentence by removing “overall” and changed it to “Structure determination and VaPomAB architecture”.

- 2. Line 87: “Intact VaPomAB is an anisotropically shaped complex and shows preferential orientation of particles in vitreous ice.” The data were not presented in the main figures or supplemental figures.*

We thank the reviewer for pointing this out. VaPomAB is indeed an anisotropically shaped protein complex, as shown in **Fig. 1a** where we annotated the three dimensions of the PomAB cryo-EM map. During the early stage of this project, our goal was to obtain the intact VaPomAB cryo-EM structure. However, we could only achieve a medium resolution of the cryo-EM reconstruction, which we speculated was due to the conformational dynamic of PomB PGB. This is why we removed the PomB PGB during the protein purification, as mentioned in the main text. We also added zwitterionic detergent CHAPSO to the purified protein sample immediately before preparing the EM grids to break the particle’s preferential orientation and to improve vitreous ice quality. These steps were critical in obtaining the high-resolution cryo-EM map of VaPomAB. In **Fig.S2c, S2d**, we present a cryo-EM image of VaPomAB without adding CHAPSO, as well as particle 2D classes and reference the figures in the main text.

- 3. Fig. 1: Cryo-EM structure at 2Å resolution should be further validated by comparing zoom-in cryo-EM density maps and atomic models, especially at the ion-binding sites. The topology diagram in panel f is not scaled appropriately.*
- 4. Similarly, it’s important to show the densities that are formed by the water molecules or ions.*

We agree with the reviewer's feedback. In **Fig. 1e**, we show the computed local resolution map, which focuses on a section near the ion binding site.

To better highlight the density coming from the ions, water molecule and the residues near the ion binding site, we have changed the color scheme of the electrostatic potential map in the revised **Fig. 3b, 3c**. Additionally, in **Fig. S2i, S2j**, we have added more panels to show zoomed-in cryo-EM density maps, overlaid on the atomic model near the ion binding sites, with different map contour levels.

To ensure proper scaling, we have modified the topology diagram in **Fig. 1f**.

5. *It might be beneficial to thoroughly compare the structures of the sodium-driven stator complex and the proton-driven stator complexes.*

It is indeed a good suggestion, which was also raised by reviewer 1 (comment #17). In **Fig. S8g-i**, we have included comparisons of the structures of the Na⁺ binding site 2 in *V. alginolyticus* PomAB, with the corresponding sites in *C. jejuni* MotAB and *B. subtilis* MotAB. The comparison clearly shows that the equivalent site in the proton-driven stator unit lacks Na⁺ contact environment and therefore does not support Na⁺ binding. Furthermore, in **Fig. 6** and **Fig. S16**, we included the T ring structure of the flagellar basal body. This structure is necessary for the stable incorporation of the sodium-driven stator unit PomAB onto the rotor, but it is not required for proton-driven stator unit.

REVIEWERS' COMMENTS

Reviewer #1 (Remarks to the Author):

I've read the responses to my questions carefully and are satisfied with the changes that the authors made.

I've also read the revised manuscript. Overall, I think this manuscript is a nice contribution and of interest for people in the field.

Reviewer #2 (Remarks to the Author):

The authors have thoroughly addressed the concerns from the reviewers. The revised manuscript is excellent. It should be accepted for publication.